# Potential Assessment of Dehydration during High-Intensity Training Using a Capacitance Sensor for Oral Mucosal Moisture: Evaluation of Elite Athletes in a Field-Based Survey

Gen Tanabe [1,2], Tetsuya Hasunuma [3,4], Yuto Inai [3,5], Yasuo Takeuchi [6], Hiroaki Kobayashi [6,7], Kairi Hayashi [1], Shintaro Shimizu [1], Nana S Kamiya [1,3], Hiroshi Churei [1], Yuka I Sumita [2], Katsuhiko Suzuki [8], Naoki Moriya [3,9] and Toshiaki Ueno [1,*]

1 Department of Sports Medicine/Dentistry, Graduate School of Medical and Dental Sciences, Tokyo Medical and Dental University, Tokyo 113-8510, Japan; gen.spmd@tmd.ac.jp (G.T.); k.hayashi.spmd@tmd.ac.jp (K.H.); sshispmd@tmd.ac.jp (S.S.); nshspmd@tmd.ac.jp (N.S.K.); chu.spmd@tmd.ac.jp (H.C.)
2 Department of Maxillofacial Prosthetics, Graduate School of Medical and Dental Sciences, Tokyo Medical and Dental University, Tokyo 113-8510, Japan; yuka.mfp@tmd.ac.jp
3 Public Interest Incorporated Association, Japan Triathlon Union, Tokyo 160-0013, Japan; hsnmtetsuya@gmail.com (T.H.); tri0410inai@gmail.com (Y.I.); moriya@jtu.or.jp (N.M.)
4 Faculty of Human Development and Culture, Fukushima University, Fukushima 960-1296, Japan
5 Graduate School of Sport Sciences, Waseda University, Saitama 359-1192, Japan
6 Department of Periodontology, Graduate School of Medical and Dental Sciences, Tokyo Medical and Dental University, Tokyo 113-8510, Japan; takeuchi.peri@tmd.ac.jp (Y.T.); h-kobayashi.peri@tmd.ac.jp (H.K.)
7 Oral Health Center, Sumitomo Corporation, Tokyo 100-8601, Japan
8 Faculty of Sport Science, Waseda University, Saitama 359-1192, Japan; katsu.suzu@waseda.jp
9 Bunka Gakuen University, Tokyo 151-8523, Japan
* Correspondence: t.ueno.spmd@tmd.ac.jp; Tel.: +81-3-5803-5867

**Abstract:** Background: The aim of this clinical study was to reveal the relationship between body dehydration and oral mucosa moisture measured by the use of a capacitance sensor for oral epithelial moisture. Methods: The following clinical parameters were recorded from each one of 19 athletes in a one-week period of high-intensity exercise at the U-23 Triathlon Training Camp in summer and winter; body weight, urine specific gravity, oral mucosa moisture, subjective oral thirst, and subjective throat thirst (within 30 min after waking and before breakfast at 7:00 a.m. on Day2 and Day6). Results: There were no significant differences in the mean values of body weight, urine specific gravity, oral mucosa moisture, oral thirst, and throat thirst between Day2 and Day6 in both measurements in summer and winter. The oral mucosa moisture had a moderate negative correlation with urine specific gravity ($p < 0.05$, r = −0.45). Conclusions: This study suggests that oral mucosal moisture determined using an oral moisture-checking device could be a potential index for assessing dehydration during sports activities.

**Keywords:** dehydration; capacitance sensor; oral mucosal moisture; urine specific gravity

## 1. Introduction

Body temperature elevations due to physical activity elicit responses of increased skin blood flow and increased sweat secretion [1]. Sweat evaporation provides a primary function of heat loss in the body, but both water and electrolytes are also lost. Water and electrolyte imbalances (dehydration and hyponatremia) can develop and adversely impact individuals' exercise performance and health [2].

Dehydration of >2% body weight (BW) can degrade aerobic exercise performance [3] and cognitive/mental performance [4]. Dehydration of over 3% BW degrades muscular strength [5]. Furthermore, dehydration impairs exercise performance and contributes to serious heat exhaustion [6] and exertional heat stroke [7], rhabdomyolysis [8], and

exercise-associated hyponatremia [9]. Dehydration can lead to death in severe cases; thus, dehydration assessment is important.

Table 1 shows an assessment of several dehydration biomarkers and subjective indicators. Total body weight and plasma osmolality are very valid and precise measures of body hydration but are of low practicality for use by most persons [2]. In physical activity, the biomarkers should be practical for use by individuals, coaches, and so on. Measures of urine biomarkers such as urine specific gravity and osmolality provide valuable insight [10], and urine color and volume are often used as subjective indicators. These urine indices are used in sports because of their practicality and especially urine specific gravity has been reported to be an invasive, inexpensive, simple, fast, and accurate indicator of hydration status before exercise [11]. Urine specific gravity is often used in clinicopathology to assess kidney function. The kidney minimizes water loss by concentrating urine. Therefore, urine specific gravity of <1.020 g/mL is indicative of dehydration.

**Table 1.** Assessment of dehydration biomarkers.

|       | Measure | Euhydration Cutoff |
|-------|---------|--------------------|
| Blood | Plasma osmolality | <290 mOsmol |
| Urine | Urine specific gravity<br>Urine osmolality<br>Urine color | <1.020 g/mL<br><700 mOsmol |
| Weight | Body weight<br>Total body water | <1%<br><2% |
| Oral | Thirst<br>Mouth dryness | |

Oral characteristics such as wrinkles and dryness of the tongue and subjective feelings of thirst in the mouth and/or throat have also been used to assess dehydration. Recent studies have reported the utility of salivary viscosity [12], saliva osmotic pressure [13], capillary refill time [14], and oral moisture [15] as diagnostic indicators.

All these indices have limitations such as the time, cost, and labor required for collection during exercise, the fact that they may be invasive and require technical expertise, and the fact that there is a time lag between the current symptoms and the indices, so there is a need for a simpler evaluation method that can provide immediate feedback.

Recently, in the field of medicine, evaluation of dehydration severity using a capacitance sensor for oral epithelial moisture has been attempted for dehydrated patients brought to emergency rooms, and it was reported that oral moisture measurement is useful for initial evaluation of dehydration [15]. Dry mouth is one of the most common clinical findings in dehydrated patients and is often associated with dehydration. McGee et al. reported that 85% of dehydrated patients experience dry mouth [16]. In recent years, the measurement of oral moisture has begun to be applied in not only the evaluation of oral dryness among elderly populations [17] but also a variety of situations in clinical practice, such as maxillofacial prosthesis patients [18,19], oropharyngeal cancer [20], xerostomia due to side effects of medication [21], and so on.

Mucus® (Life Co., Saitama, Japan; Figure 1), a capacitance sensor for oral epithelial moisture, easily measures oral moisture within two seconds [22]. This device is the first to measure oral moisture, and the principle of this device is that the epithelial moisture is measured as capacitance. The dielectric constant of water is much higher than that of other substances; therefore, the percentage of water in the substance can be checked by measuring the dielectric constant of the substance. The moisture of the substance is measured by calculating the capacitance from the dielectric constant of the substance, and the principle is the same as that of a skin moisture sensor. The indicated value (%) of an oral moisture meter is based on the gravimetric moisture content of a standard sample of a special protein membrane for medical use and is expressed as a percentage of moisture

content. The definition of moisture content by the gravimetric method is the following: moisture content = $B/(A + B) \times 100\%$ (A: weight of dry protein membrane, B: weight of water), where 100% represents the state when the sample is entirely water. However, the measurement range of the moisture meter is about 15 to 65%. The reliability of the data was previously demonstrated by comparison with the dry weight method [22]. Fukushima et al. reported that the sensitivity and specificity values are close to 80%. The oral moisture values range from 0 to 99.9, and values of $\geq$29.6, 28.0–29.5, and $\leq$27.9 are defined as normal, borderline dry mouth, and dry mouth, respectively [23]. Takano et al. reported that using an oral moisture-checking device has sufficient intra- and inter-investigator reliabilities [24]. As such, Mucus® has received manufacturing and marketing approval as a body composition analyzer (approval number: 22200BZX00640000) by the Pharmaceutical and Medical Devices Agency of Japan.

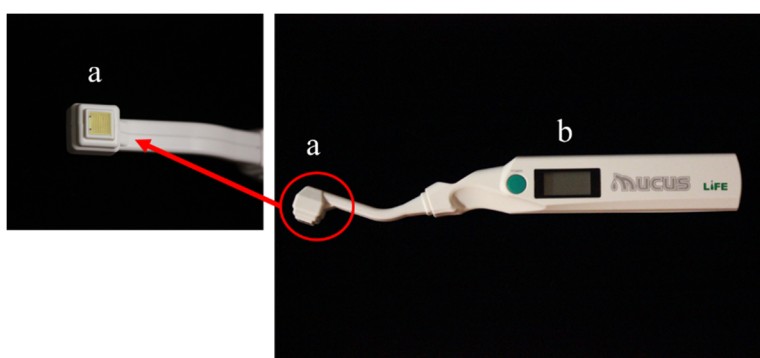

**Figure 1.** Mucus®; Life Co., Ltd., Saitama, Japan, (**a**) is a capacitance sensor; (**b**) is the score display.

We hypothesized that oral moisture measures by the use of such a device may provide a rapid and precise indication of dehydration in sports. The aim of this clinical study conducted in a one-week period of high-intensity exercise at the U-23 Triathlon Training Camp during summer and winter 2019 was to reveal the relationship between body dehydration and the oral mucosa moisture measured by the use of a capacitance sensor for oral epithelial moisture.

## 2. Materials and Methods

### 2.1. Study Subjects

This study was approved by the research ethics committee of Tokyo Medical and Dental University (approval number: D2019-031). Before the start of this study, all the study participants received verbal explanations regarding data collection and the protection of privacy and personal information, and they each signed an informed consent form.

Data were collected at the U-23 Triathlon Training Camp during summer and winter 2019. Samples of 6 subjects in the summer camp and 13 subjects in the winter camp were recruited from the participants of the U-23 Triathlon Training Camp; those diagnosed with oral mucosal abnormalities of the tongue and/or Sjögren's syndrome or xerostomia were not invited to participate in this study.

### 2.2. Data Collection

The camp lasted for seven days, of which the mornings of the second and sixth days were set as measurement days. Within 30 min after waking and before breakfast at 7:00 a.m., data regarding body weight, urine specific gravity, oral mucosa moisture, oral thirst, and throat thirst were collected from each subject. Body weight was measured and recorded on a scale (YAMAZEN HCF-40 Life Co., Ltd., Saitama, Japan). Urine specific gravity was measured by collecting urine early in the morning and measuring it with a urine specific gravity meter (PEN urine specific gravity meter Atago Co., Ltd., Tokyo, Japan). The extent of oral mucosa moisture was measured using a capacitance sensor

for oral epithelial moisture (Mucus®; Life Co., Ltd., Saitama, Japan). It was measured on the center of the lingual mucosa approximately 10 mm from the tip of the tongue; the device was manually applied at a pressure of approximately 200 g by a single measurer (Figure 2). A dedicated disposable polyethylene cover was applied to the sensor for each subject. The measurements were done three times, and the median values were recorded. Subjective scores of oral and throat thirst were recorded on an 11-point scale from 0 to 10 (10: strong sense of thirst). Furthermore, the weather and maximum temperatures during the camp were recorded, and the session rating of perceived exertion (s-RPE) was recorded as the exercise intensity during the camp. For the measurement of the s-RPE, the modified Borg CR-10 scale was used to quantify session intensity [25]. Following the final training session each day, each participant individually recorded the subjective exercise intensity for that day.

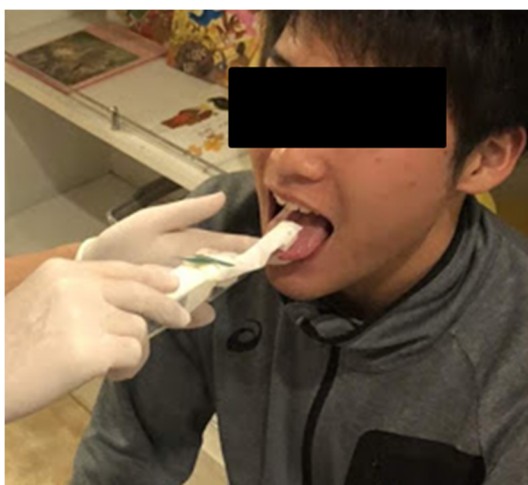

**Figure 2.** An example of measurement of oral mucosal moisture by using an oral moisture checking device (Mucus®; Life Co., Ltd., Saitama, Japan).

### 2.3. Data Analysis

Data regarding body weight, urine specific gravity, oral mucosa moisture, oral thirst, and throat thirst were statistically analyzed using Student's paired *t*-test to compare the second day with the sixth day. The correlations of the oral mucosa moisture and urine specific gravity results were examined via Spearman's correlation coefficient, and the regression line was calculated. The statistical analyses were performed using statistical software JMP14 (SAS Institute Inc., Cary, NC, USA) at a 5% significance level.

### 3. Results

The study participants' demographic data (sex, age, body weight) are summarized in Table 2. The weather and maximum temperature during the camp is shown in Table 3. The summer camp had mostly sunny and hot days over 30 °C, while the winter camp had many rainy, cloudy, and cold days. Transitions in body weight, urine specific gravity, oral mucosa moisture, oral thirst, and throat thirst in the summer and winter camps are shown in Figure 3. There were no significant differences in the mean values of body weight, urine specific gravity, oral mucosa moisture, oral thirst, and throat thirst between Day2 and Day6 in both measurements in summer and winter. In the summer, the urine specific gravity was above 1.020 g/mL, which is considered in sports to indicate dehydration, and the oral mucosa moisture measured by the Mucus® device was below 28.0, which is diagnosed as xerostomia. In the summer, all values tended to worsen at the end of the camp, but in the winter, they tended to improve because summer is the heat and humidity environment [26].

**Table 2.** Subject data, mean ± SD.

| Number of subjects | 19 (Male 10; Female 9) |
| --- | --- |
| Mean age | 20.8 ± 0.8 |
| Weight (kg) | 58.1 ± 8.7 |

**Table 3.** The weather and maximum temperatures during the summer/winter camps.

| Summer | Day1 | Day2 | Day3 | Day4 | Day5 | Day6 | Day7 |
| --- | --- | --- | --- | --- | --- | --- | --- |
| Weather | Sunny | Sunny | Sunny | Sunny | Sunny | Sunny | Sunny |
| Maximum temperature (°C) | 30.1 | 30.3 | 30.5 | 33.5 | 32.6 | 31.1 | 32.0 |
| s-RPE [1] | 6.6 ± 2.7 * | 34.0 ± 10.5 | 25.5 ± 7.4 | 13.3 ± 4.5 | 33.0 ± 10.5 | 39.4 ± 8.8 | 10.3 ± 2.9 |
| **Winter** | **Day1** | **Day2** | **Day3** | **Day4** | **Day5** | **Day6** | **Day7** |
| Weather | Rain | Rain | Sunny | Cloudy | Rain | Rain | Sunny |
| Maximum temperature (°C) | 13.2 | 10.2 | 17.1 | 15.8 | 16.5 | 16.4 | 12.7 |
| s-RPE | 2.3 ± 3.60 | 9.9 ± 3.7 | 32.8 ± 12.3 | 17.6 ± 4.2 | 24.5 ± 11.3 | 20.7 ± 6.6 | 24.6 ± 6.7 |

[1]: s-RPE is the session rating of perceived exertion. *: Average ± S.D.

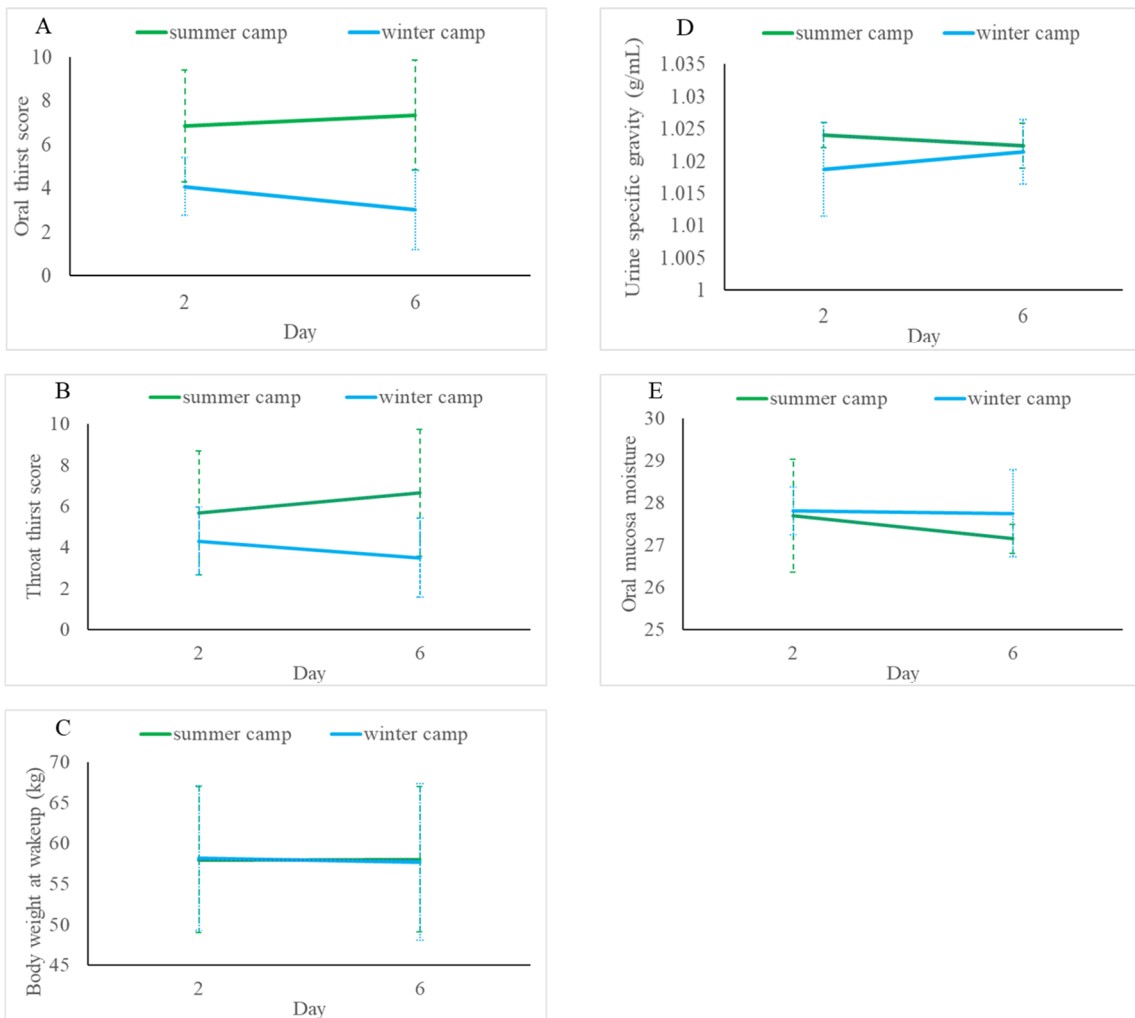

**Figure 3.** Transitions in oral thirst (**A**), throat thirst (**B**), body weight at wakeup (**C**), urine specific gravity (**D**), and oral mucosa moisture (**E**) during the summer and winter camps. Urine specific gravity above 1.020 g/mL indicates dehydration in sports. Oral moisture values range from 0 to 99.9, and values of ≥29.6, 28.0–29.5, and ≤27.9 are defined as normal, borderline dry mouth, and dry mouth, respectively. Error bars reflect standard deviation.

Meanwhile, the Spearman's correlation coefficient illustrated that the oral mucosa moisture had a moderate negative correlation with urine specific gravity ($p < 0.05$, r = −0.45), and the regression equation showed that the value of oral mucosa moisture predicting urine specific gravity of 1.020 g/mL, which is an index of dehydration, was ≈27.5 (Figure 4). It was analyzed the multivariate correlations among body weight, urine specific gravity, oral mucosa moisture, subjective oral thirst, and subjective throat thirst, but results correlating with oral mucosal moisture were not observed, except for urine specific gravity.

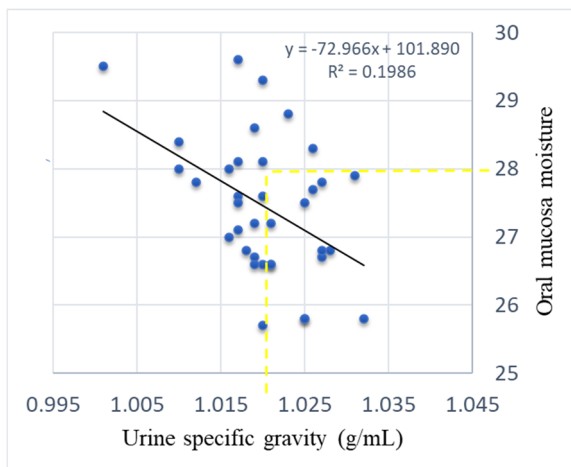

**Figure 4.** A scatter plot of the oral mucosa moisture and urine specific gravity for both the summer and winter camps. The oral mucosa moisture had a moderate negative correlation with urine specific gravity ($p < 0.05$, r = −0.45). The regression equation showed that the value of oral mucosa moisture predicting urine specific gravity of 1.020 g/mL, which is an index of dehydration, was ≈27.5.

## 4. Discussion

The findings of this study suggest that oral mucosa moisture measured using a capacitance sensor for oral epithelial moisture is a simple index for dehydration assessment during sports activities. Although no subjects were diagnosed with heat stroke in either camp, the urine specific gravity of the subjects ranged from 1.001 to 1.032 g/mL, which means that the subjects included both normal and possibly dehydrated conditions. These results suggest that the subjects in this study were appropriate subjects for the evaluation of dehydration during high-intensity training periods.

For these subjects, a moderate negative correlation of the oral mucosa moisture and urine specific gravity is shown in Figure 4. The results agree with the finding reported by Fukushima et al. that there is a negative correlation between the severity of dehydration and the degree of oral mucosal moisture ($p < 0.05$, r = −0.69). The index of mouth and throat thirst which has been used as an indicator of dehydration is a subjective score of the athlete themself. The fact that oral mucosal wetness is numerical and objective data, and its value is correlated with urine specific gravity is expected to have a great impact on future dehydration indicators. Although the correlation between the oral mucosa moisture and urine specific gravity was an important finding, the following points should be noted. As shown in the scatter plot, there were nine cases in which the urine specific gravity was less than 1.020, but the oral wetness was plotted lower than 28.0. In this case, the urine specific gravity may have failed to assess the state of dehydration, or the oral mucosal wetness may have overestimated, but this cannot be determined from this study design alone. Furthermore, there were two cases where the urine specific gravity was 1.020 or higher, but the oral wetness was higher than 28.0. These results suggest that the oral mucosa moisture measured using a capacitance sensor for oral epithelial moisture as a simple index for dehydration assessment on its own may not be sufficient, and combining multiple assessment methods may be necessary. Simplified approach to assessing daily dehydration is using a Venn diagram decision tool [27]. Weight loss (>1%), dark-colored

urine (>5 arbitrary units, a.u), and thirst (conscious desire for water) are used as markers of inadequate fluid intake. In addition, it may be useful to continue to take daily timed measurements within the same athlete to capture individual data and compare it to the data in the field dehydration index.

The regression equation in Figure 4 shows that the value of oral mucosa moisture predicting urine specific gravity of 1.020 g/mL, which is an index of dehydration, was $\approx$27.5. Since this value is less than 28.0, which is diagnosed as xerostomia, the cutoff line for dehydration cannot be determined in this study design, but it is expected to be in a reasonable range. Epidemiological studies to examine the cutoff line for dehydration determined via oral mucosa moisture need to be done in the future. The Mucus® device is intended to measure oral mucosa moisture in elderly patients with xerostomia. Therefore, it may be possible to modify this measurement for dehydration by changing the dielectric ratio.

Although there are multiple measures of dehydration, it is necessary to consider the validated targets, in other words, acute dehydration index or chronic dehydration index. TBW and plasma osmolarity are indicators of acute and chronic dehydration, while urine specific gravity and urine osmolarity are indicators of chronic dehydration [2]. The oral mucosa moisture was previously compared with acute indications [15,28]. Suzuki et al. examined changes in oral mucosal moisture during a 60-min bike ride in the laboratory at temperatures of 15, 23, and 30 °C. The results showed a gradual downward trend in oral mucosal moisture at all temperatures, with a smaller value recorded at 30 °C throughout the 60 min [28]. In this study, a correlation was confirmed with urine specific gravity, which is an indicator of chronic dehydration. This suggests that it may be possible to evaluate not only acute dehydration in emergency situations but also chronic dehydration that can be applied in daily condition management.

The capacitance sensor for oral epithelial moisture, Mucus®, employed in this study currently has no restrictions on who can use it, and it can be easily employed by anyone. In addition, it is not large and can be carried easily. The measurement time is only a few seconds, which is less of a burden for the athlete and the measurer. However, when we performed measurements with it in this study, we found that it was somewhat technically sensitive in terms of the pressure and angle of the contact surface, and in some cases, practice was necessary before use. The design makes it difficult to measure by oneself. There is room for improvement in the design to make it more universal, so that it can be used by anyone, anywhere, at any time.

The design of this study was a field survey, and environmental factors and levels of exercise load that influence dehydration were recorded but not controlled. Comparing their oral mucosa moisture measurement with other gold standard methods such as salivary flow rate is necessary. The relationship with other indicators of dehydration, such as plasma osmolality, also needs to be evaluated.

## 5. Conclusions

This study indicated that oral mucosal moisture determined using an oral moisture-checking device could be a simple valuable index for assessing dehydration during sports activities.

**Author Contributions:** Conceptualization, G.T. and T.U.; methodology, G.T.; validation, Y.T. and H.K.; formal analysis, G.T.; investigation, G.T., T.H., Y.I., Y.T., H.K., K.H., S.S., N.S.K. and H.C.; data curation, G.T.; writing—original draft preparation, G.T.; writing—review and editing, Y.I.S. and T.U.; supervision, T.H., K.S., N.M. and T.U.; project administration, H.C.; funding acquisition, G.T. All authors have read and agreed to the published version of the manuscript.

**Funding:** This research was funded by JSPS KAKENHI, Grant Number 20 K18824, and by YAMAHA Motor Foundation for Sports, Grant Number 2983.

**Institutional Review Board Statement:** The study was conducted according to the Declaration of Helsinki and approved by the Institutional Review Board (or Ethics Committee) of the research ethics committee of Tokyo Medical and Dental University (approval number: D2019-031).

**Informed Consent Statement:** Informed consent was obtained from all subjects involved in the study.

**Data Availability Statement:** The data presented in this study are available on request from the corresponding author. The data are not publicly available due to elite athlete.

**Conflicts of Interest:** The authors declare no conflict of interest. The funders had no role in the design of the study; in the collection, analyses, or interpretation of data; in the writing of the manuscript; or in the decision to publish the results.

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
