# Peer review of "Potential Assessment of Dehydration during High-Intensity Training Using a Capacitance Sensor for Oral Mucosal Moisture: Evaluation of Elite Athletes in a Field-Based Survey"

_chemosensors, doi:10.3390/chemosensors9080196_

Round 1

Reviewer 1 Report

The paper describes a clinical study to reveal the relationship between body dehydration and oral mucosa moisture measured by the use of a capacitance sensor. The clinical study is based on one week of 19 athletes with high-intensity exercise.

The paper is well structured and clearly written. The experiment setup and method are explained in enough detail.

The reviewer has the following comment and remarks:

  1. It will be beneficial to explain the relationship between the capacitive sensor and Oral moisture values as the paper is titled a capacitance sensor for oral mucosal moisture.
  2. It will be good to also include the number of tested athletes in the abstract
  3. The results only shown oral mucosal moisture vs. urine specific gravity, not the other parameters measured and shown in Figure.2, please elaborate?
  4. Although a good discussion section is given, the claims that there is a correlation between oral mucosal moisture and urine specific gravity seems weak, as shown in Figure 3, the R^2 is only 0.2 and the data seems to be scattered.

The reviewer thinks although the method has its place to investigate dehydration, the measured data for each athlete is too limited to give instructive results and requires further investigation.

Based on the results, the reviewer thinks the conclusion needs to indicate that the method needs further investigation and revises the conclusion sentence as ‘’oral mucosal moisture determined using an oral moisture-checking device could be a simple valuable index…’’.

Author Response

Thank you very much for reviewing our manuscript and for your valuable comments. We have read them carefully, and we have rewritten the manuscript based on them.

Reviewer 2 Report

Tanabe et al. reported dehydration measurement of athletes during high intensity training with the capacitance sensor, Mucus. They analyzed time-dependent dehydration and showed the time-dependent change in oral mucosa moisture. They also compared this data with other dehydration indicating markers such as oral thirst score, throat thirst score and urine specific gravity. Authors showed interesting data which would be helpful for the readers in this field. However, following point should be addressed for the publication.

  1. Detail explanation about urine specific gravity is necessary for the readers.

  1. Authors claimed that correlation between the oral mucosa moisture and urine specific gravity is additional important finding in this work. However, authors did not explain why that is important. Additional explanation about the importance of this finding is necessary.

  1. Higher resolution of Fig 1 is need for the publication.

  1. Picture of Mucus device is need for the readers.

  1. It would be great if authors can compare their oral mucosa moisture measurement with other gold standard methods.

  1. It would be better to describe the rationale why there is a difference between summer and winter camp.

Author Response

(The authors gave the same response as above.)
